# A Study on Hybrid Sensor Technology in Winter Road Assessment

**Aleksander Pedersen ***[ID] **and Tanita F. Brustad**[ID]

Faculty of Engineering Science and Technology, UiT The Arctic University of Norway, Lodve Langesgate 2, 8514 Narvik, Norway; tanita.f.brustad@uit.no

\* Correspondence: aleksander.pedersen@uit.no

**Abstract:** Road conditions during the winter months in Nordic countries can be highly unstable. Slippery roads combined with heavy haul traffic and ordinary road users can create dangerous, even lethal, situations if road maintenance is unsuccessful. Accidents and critical road conditions may lead to blocked roads, putting strain on a limited number of main roads in many regions, and may in the worst case isolate areas entirely. Using sensors in winter road assessment has been a popular topic for over 20 years. However, with today's developments connected to smaller and cheaper sensors, new opportunities are presenting themselves. In this study, we performed preliminary experiments on a variety of sensors, both commercial and experimental, to evaluate their benefits in possible hybrid sensor technology, which can give a more complete characterization of the road surface than what is possible from just one sensor. From the collected data and visual analysis of the results, the idea of a hybrid sensor seems promising when considering the differences in the tested sensors and how they may complement each other.

**Keywords:** sensor testing; winter road maintenance; hybrid sensor technology; risk assessment

---

## 1. Introduction

Transportation of raw material, goods, and people is a constantly growing factor in the Nordic countries. With this growth, a significant increase in semitrailers for long haul has been observed where research states that many of them are a safety risk on slippery winter roads [1]. To meet the current and future industry with accessible and, more importantly, safe roads, the pressure on successful winter road maintenance is high. Research shows that winter road maintenance is crucial to maintain safe roads [2]. Knowing when and where critical conditions occur, improving risk assessment, can increase the efficiency of maintenance personnel (saving money), and can be used to warn road users of potential risks (increasing safety). The limited number of main roads, combined with long distances between key cities in the Nordic countries, make the road network vulnerable. Critical road conditions can lead to closed roads, which again can lead to areas being isolated over prolonged periods of time. With today's developments connected to faster transfer speed of data, smaller and cheaper sensors, and a focus on smart environments, new opportunities are presenting themselves in winter road maintenance. The new popular area within the winter road assessment research field is that of hybrid measurements with transmission of information to neighbouring vehicles or to road side servers [3,4]. Hybrid measurements are becoming more and more recognized as a method that can give better overall information on road conditions, often with already available data, compared to lone sensors. This can be seen in recent research, e.g., in [5], where in-car sensor data are incorporated into weather forecast models to improve road weather forecasts; in [6], where a neural network is employed to develop a surface friction prediction model based on historic data from an optical sensor; or, in [7],

where acoustic measurements and machine learning algorithms are combined to classify dry, damp, and wet road surfaces.

The research and experiments performed in this study are a part of the Winter Road Maintenance project (WiRMa project). In the project, partners from Norway, Sweden, and Finland conduct research and demonstrate profitable, industrial, and network based systems for decision support in winter road maintenance. The project vision is to demonstrate a viable winter road maintenance decision-support system (DSS) using novel technologies for ambitious ideas by:

- combining measurement techniques for improved icing measurements;
- collecting, communicating, and crunching road weather, vibration, and road surface condition; (snow, ice, wet, slush, etc.) and indirect friction information from vehicles;
- improving forecasts and nowcasts based on vehicle observations; and
- visualizing the information to be used as a DSS in winter road maintenance.

The specific motivation surrounding this work is connected to one of the work packages in the project: Work Package 4—Winter road condition information. The overall aim of the work package is to investigate the possibility of combining new direct and indirect sensing methods together with known optical sensors to provide a more complete characterization of the road surface (hybrid measurement techniques) by turning vehicles into mobile monitoring stations. Currently, such development is on-going in the automotive industry, but that industry is very closed and the data gathered from modern vehicles are not available for winter road maintenance purposes. Additionally, the data available from vehicles are not based on continuous measurements at the moment, as, for example, slipperiness information is available only when ABS, ESP, or similar are engaged. Mobile systems with continuous measurements will play an important part in monitoring road conditions in Nordic countries, tackling problems related to making rightly timed road infrastructure maintenance decisions. As of now there are simply not enough data and information available to properly handle maintenance on roads that are subjects to harsh conditions and long distances.

Based on the motivations and aims of the WiRMa project, the aim of the study was to select and test a number of sensors to look at the possibility of equipping vehicles with hybrid measurement technology, in order to improve the risk assessment of roads in arctic conditions. The motivation behind this is to increase safety and accessibility for road users during the unpredictable winter months, and help maintenance personnel assess maintenance needs. As a final comment, it is important to remember that sensors will always only be a tool in decision support but ultimately the safety depends on the drivers' behavior of perceiving the sensor alerts, since this is considered to be one of the influential factors in reducing accidents [8]. In this paper, the process and initial results of using certain sensors in winter road assessment are described. The process includes the choices made in selecting sensors (both commercial and experimental), methods for collecting and storing data, and how the measurements were analyzed. The presented results are our initial experience with measurement data and the sensor's behavior, including a discussion where the data are analyzed and compared, in relation to reliability, similarity, and usefulness.

## 2. Preliminaries

This section gives a preliminary overview of some of the commercial sensors known in winter road assessment, and gives examples of the possibilities connected to experimental sensors within the field.

### 2.1. Commercial Sensors

When it comes to sensors created for winter road assessment, two categories exist: stationary and mobile. In this study, the focus was on mobile sensors and four optical sensors which were considered: The Road Condition Monitor 411 (RCM411) [9], Mobile Advanced Road Weather Information Sensor (MARWIS) [10], MetRoadMobile [11], and Road Eye [12]. These sensors are measuring the reflected light, from a light source illuminating the road, and analyze the data to classify the surface conditions.

The directorate of public roads [13] is also using mobile sensors, but with different measuring types: retardation (deceleration) meter, continuous friction measurement using a mobile wagon attached to the car with either one or two friction wheels, or handheld friction meters. The friction wheels used in Norway are produced by ViaFriction [14], and the directorate of public roads uses two different types, the Road analyser and recorder 5 (RoAR5) [15] and OSCAR [16]. OSCAR is the most reliable of all the friction meters used by the directorate of public roads.

### 2.2. Experimental Sensors

In addition to the sensors that have been specifically created for winter road assessment, experimental sensors, including cameras, have dominated the research field for many years. Four examples showing the variety of experimental sensor types during the last three years (2017–2019) follow. In [17], Pan et al. applied a pre-trained convolutional neural network to automatically detect road surface conditions based on camera images. Hou et al. created a system called VehSense [18], which combined a smartphone and On-Board Diagnostics (OBD-II) data to detect vehicle skidding. Microphones together with neural networks were presented in [19] as a way of detecting road surface wetness. Gui et al. [20] combined piezoelectric and optical sensors to classify road conditions and measure ice/water thickness. These four examples express only a fragment of the creativity and possibilities within the field of road condition assessment (see [21–24] for additional examples).

### 3. Method

This section describes the choice of sensors used in the experiments, both commercial and experimental. In addition, the placement of the sensors on the vehicle is shown, an overview of how the data for each sensor were collected and stored is given, and lastly how the data were evaluated is explained.

### 3.1. Choice of Commercial Sensors

Based on the project description, one of the key aspects was to assess road condition using measurement techniques. Initially, when we became part of the project, two sensors were already bought and ready to use: RCM411 and MARWIS. The next step was to assemble and test these on a vehicle. Previously, we were discussing four different optical sensors, and since two sensors were already at our disposal and we know that they are commercially available, with extensive support, there was no need to buy any new equipment initially. In addition, the Road Eye sensor (Prototype, Optical Sensors, Gothenburg, Sweden) was to be used in another work package in the project, thus naturally two different sensors were chosen for this work package. Another reason for choosing RCM411 (411, Teconer, Helsinki, Finland) and MARWIS (8900.U04,Lufft, Fellbach, Germany) was because they measure a range of variables, unlike Road Eye and MetRoadMobile (NA, MetSense, Gothenburg, Sweden) that only measure one and two variables, respectively. Both RCM411 and MARWIS are used specifically for measurements on roads, and the directorate of public roads have tested them. As a result, these two commercial sensors are in a much higher price range than the experimental sensors, but then more reliable and suited for their specific purpose.

### 3.2. Choice of Experimental Sensors

The main attributes, in our opinion, which the experimental sensors should possess are: a low price, a reasonable size, should complement the commercial sensor in a beneficial way, and, preferably, be available on the market in some form. Taking these attributes into account, as well as the research done in the literature cited in Section 2.2, the main experimental sensor of choice ended up being a Walabot radar [25], with the supplementary selections of vehicle data (OBD data), video, sound, and smartphone sensors.

The Walabot radar is a pocket-sized 3D imaging sensor that uses radio frequency technology to illuminate the area in front of it and sense the returning signals. The sensor supports short range

scanning and distance scanning, with the possibility to extract 3D image data, 2D image slices, raw signals, and the sum of raw signals in an image (image pixel energy).

The reason for choosing the Walabot sensor is its small size and low cost, as well as it being a radar. This is of interest since radar technology has not been tested extensively for winter road assessment, which in theory should be suited for given that radar is not dependent on sight, being less affected by snowy or foggy conditions where, for instance, optical sensors have a problem. Thus, since the commercial sensors of choice were both optical sensors, a radar sensor should compliment these sensors in a beneficial way. The other supplementary selections were chosen because of the simplicity to obtain the data, and because they give a variety of information that the other sensors do not give. Vehicle data are already measured by the car and easily acquired by an OBD adapter (we chose a second generation adapter, OBD-II [26,27]). As for the other data, most people today carry around a smartphone that can both record sound and video, in addition to having a number of other incorporated sensors.

### 3.3. Sensor Placement on the Vehicle

Figure 1 shows the placement of the four main sensors (RCM411, MARWIS, Walabot, and OBD-II) on the car. The RCM411 sensor was mounted on the tow ball pointing at the right wheel track, the MARWIS sensor was secured with a rack to the truck bed pointing at the center of the road behind the vehicle, the Walabot sensor was attached below the back left passenger door pointing straight down at the wheel track, and the OBD-II (Model MX 201, OBD Solutions, Phoenix, AZ, USA) adapter was plugged into the adapter inside the car. All these sensors had their fixed placement throughout the study, except for the Walabot (Developer Kit, Vayyar Imaging Ltd., Fairfield, OH, USA), where various placements were considered before choosing the above described place. The sensors not mentioned here were not part of the main setup, but rather mounted and tested on small intervals under chosen conditions. This included a microphone attached at various places in the dashboard, and a smart phone attached to a phone mount on the dashboard used as a dashcam to capture video of the road in front (which included sound from inside the car). In addition, the dashcam phone also utilized several in house applications for different sensor measurements.

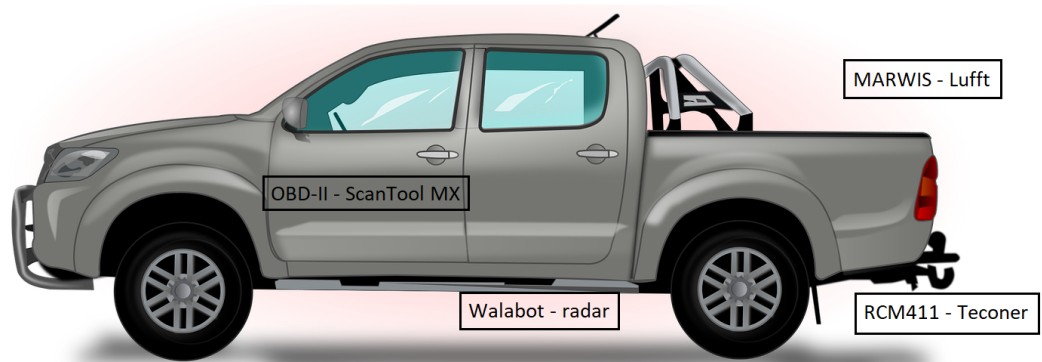

**Figure 1.** Placement of the sensors on the car. Image by OpenClipart-Vectors from Pixabay.

### 3.4. Data Collection and Storage

The data collection and storage were different for each individual sensor. Next follows a description on the methods used, for each sensor, for collecting and storing data. As an overall storage place for all sensors data in all experiments, to ensure easy access and sharing, Box [28] was used.

The RCM411 sensor transferred data to a mobile phone using Bluetooth, which then communicated them to selected servers created specifically for RCM411 sensor data. The servers make it possible to see real-time updates on surface conditions online at https://roadweather.online, and download data (in comma-separated values files) from the sensors one owns. Since the online

application already existed, was simple to use, and stored the data intuitively, this became the method used for collecting and storing data. In addition, the files from https://roadweather.online were downloaded and stored in Box.

The MARWIS sensor was compatible with two existing smartphone applications, were one had equivalent functions as the RCM411 application, the MARWIS application. This app, similar to the RCM411 application, had no internal storage capability and only an option to communicate with a remote server, which required login information which we opted not to use since downloading of raw data seemed impossible. Using the browser application ViewMondo (2.11.0, Informatik Werkstatt GmbH, Augsburg, Germany) the data from MARWIS are visually good, but there were no available tools for converting the processed data back into a raw format. The other application, ConfigTool (1.1.947, Lufft), was a configuration tool which included a collection of the different sensor variables with an associated visualization. ConfigTool also had an option to save the data to an xml file in the internal storage. Since this was the only suitable option, initially, we decided to store the data in this format. All the measurement data from the sensor were communicated to the phone using Bluetooth, and the ConfigTool files were uploaded to Box.

The Walabot sensor was run from a Raspberry Pi where data were collected and stored, initially. The Raspberry Pi ran a C++ program that collected 4–5 measurement every second from the Walabot, and saved them in comma-separated values files (csv files). To prevent large data loss, in the case of an interrupted connection, a new file was created after a given time interval, storing the data in multiple files instead of just one. Then, after ending a test-run, the data were assembled into one file and stored in Box.

The vehicle data (OBD data) were collected from the vehicle through an OBD adapter (OBD-II) that extracted data from the vehicle and made them available via Bluetooth on a smartphone app. The app lets one select which variables OBD-II should extract from the vehicle, where the possible choices depend on the model and type of vehicle involved. From the app, csv files were available for download, and the downloaded files were then stored in Box.

### 3.5. Measurement Analysis

To evaluate the sensors, in a winter road assessment perspective, the following characteristics were examined and analyzed: possible data from the sensors, reliability of the sensors (including existing software), collection and storage experience, and test-run results. An overview of the evaluation criteria is given in Table 1. Example data from each sensor are presented and discussed with a focus on relevance in winter road assessment. The reliability of the sensors, including any available software, was considered and evaluated based on our experience when testing them. Both the reliability and experiments presented in this paper are preliminary results recorded when exploring and configuring this type of hybrid setup with multiple sensors and devices being run simultaneously. Collection and storage methods, chosen above, were analyzed in regards to fulfilling the task we initially needed them to fulfill. Lastly, the initial test-run results were analyzed with visual comparison by observing the differences of two graphs against each other. For the RCM411 sensor, a consistency test was performed to evaluate the stability of the sensor, while, for MARWIS and Walabot, a comparison against RCM411 data was made.

**Table 1.** Description of the evaluation criteria.

| Evaluation Criteria | | Criteria Description |
|---|---|---|
| Sensor variables: | S1. | Number of relevant variables |
| | S2. | Variable relevance (High, Medium, Low, Exp (Experimental)) |
| Reliability: | R1. | Reliability in the initial connection (High, Medium, Low) |
| | R2. | Reliability during measurements (High, Medium, Low) |
| | R3. | Reliability of the software (High, Medium, Low) |
| Data collection and storage: | D1. | Format of the collected data |
| | D2. | Assembly of files required (Yes, No, On stop (If the car power is turned off)) |
| | D3. | Storage (Local, Online, Both) |
| Test-run results | T1. | Initial conclusion of the comparisons |

## 4. Results and Discussion

This section includes the results of the initial tests of the three sensors, namely RCM411, MARWIS, and Walabot, with a discussion connected to the evaluation methods described in Section 3.5. In addition, the supplementary sensors are considered with a focus on our experience when using them, and a possible hybrid technology is discussed.

### 4.1. RCM411

This section describes the sensor variables available from the RCM411 sensor, gives a discussion on the reliability of the sensor based on our experience, and comments on collection and storage of the data. In addition, a consistency test of the variables friction, surface temperature, and road state (defined here as road surface condition) is shown.

#### 4.1.1. Sensor Variables

The RCM411 sensor offers a range of different variables from the measurements. In the case of winter road assessment, the most interesting are: friction, road state, air temperature, surface temperature, and water layer thickness. For the road state, a number in the range 1–6 is given where 1 = dry, 2 = moist, 3 = wet, 4 = slush, 5 = ice, and 6 = snow. Figure 2 shows three examples of extracted data from the RCM411.

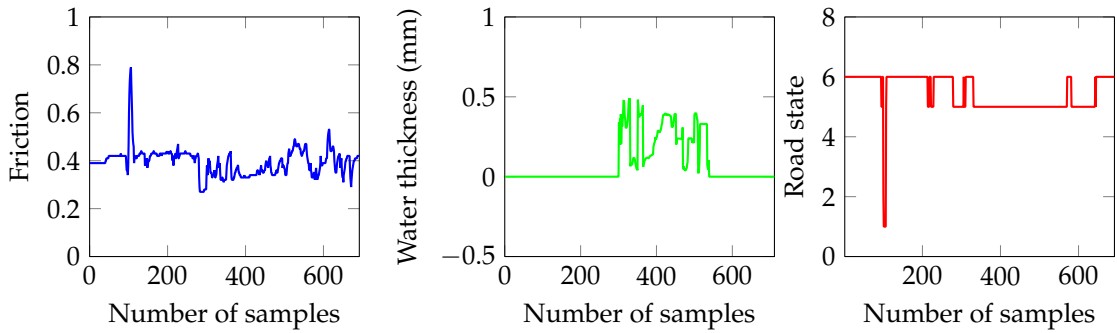

**Figure 2.** Three of the possible data types that can be extracted from the RCM411: (**left**) friction; (**middle**) water thickness; and (**right**) road state.

#### 4.1.2. Reliability

The RCM411 sensor was one of the most reliable sensors, in this initial study, in regards to uptime. Through the experiments, the sensor did not once experience disconnection problems or lose data during a run. The sensor was easy to mount on the vehicle and establishing a connection against a smartphone went painlessly. To us, the online resource showing real-time updates is dependable and useful, expressing the variables friction, water thickness, air temperature, and road temperature as a graph, and road state by adding color to the traced path.

#### 4.1.3. Data Collection and Storage

The data collection and storage from the RCM411 sensor, via the online web-page, worked. One csv file was created for each run, with measurements taken once every second, and, by logging into our account, the files, sorted by date and time, were easily downloaded. Since there was only one measurement per second from the sensor during the test-runs, the created files were not overly large and well suited for analysis, even for runs lasting more than an hour.

#### 4.1.4. Consistency Test

The initial test-run of the RCM411 sensor was a consistency test where a decided track was driven twice in order to observe the similarities between the two runs. A comparison was made between three variables, namely friction, surface temperature, and road state, to evaluate the consistency of

the measurements. The plots of the frictions can be seen in Figure 3, surface temperatures in Figure 4, and road states in Figure 5. The test-route, located in Narvik, Norway, had a length of about 5 km and took 12 min to drive through, and the weather conditions were sunny with an average air temperature of −4 degrees Celsius. In the comparisons, we do not expect to see two equal graphs for the measured variable, because we know that conditions were not exactly the same for the two runs. The influence of other road users was present, as well as the impact from the sun, and the varying speed of the vehicle. However, we do expect to see similarities between the graphs in some areas.

In the figures, it can be observed that there are similarities between the variables when comparing the two runs. Especially in the friction and road state plots where the runs give equal values for the variables in certain areas, more so for the state than the friction, which is reasonable given that the friction can change within a road state. In the plot of the surface temperature (Figure 4), the observation is that they have the same behavior but the values of the second run are consistently higher than for the first run. This can be explained by the sunny conditions where the air temperature was on average one degree Celsius higher during the second run compared to the first run. An overall conclusion to the consistency test is that the sensor appears to be stable when measuring the variables friction, surface temperature, and road state.

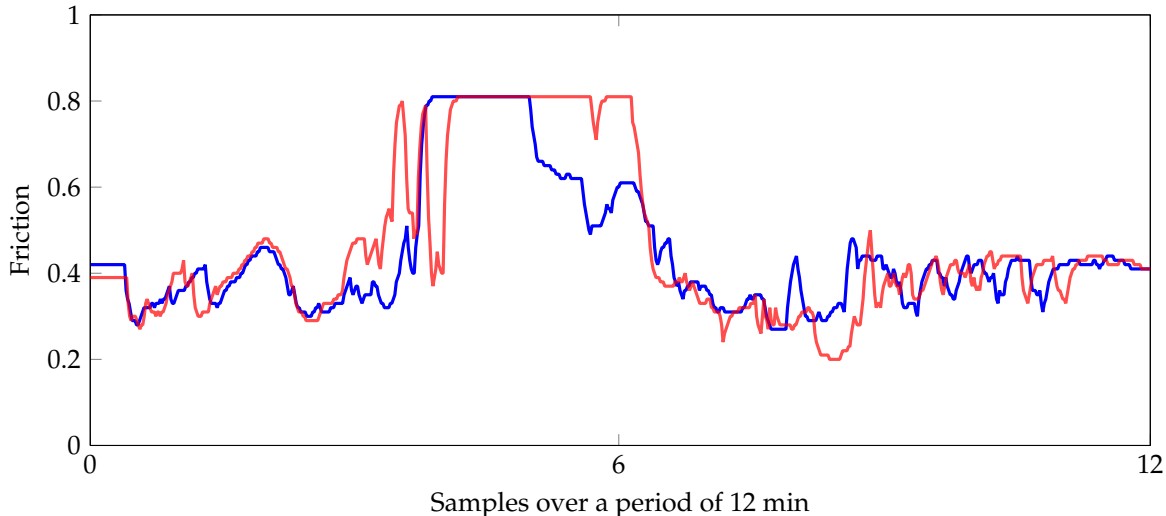

**Figure 3.** RCM411 friction plots from two runs along the same test route: blue, run 1; red, run 2.

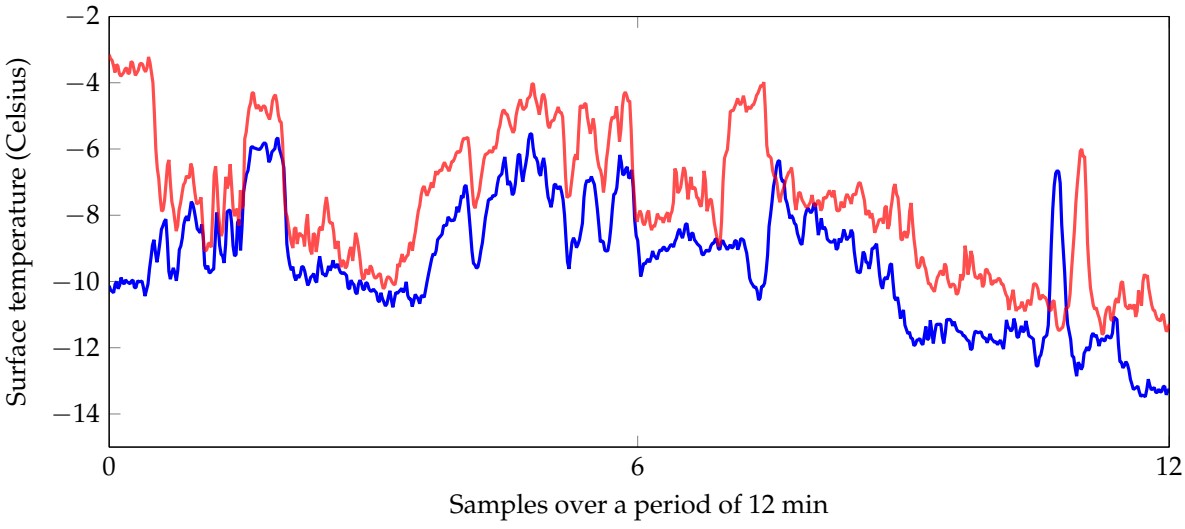

**Figure 4.** RCM411 surface temperature plots from two runs along the same test route: blue, run 1; red, run 2.

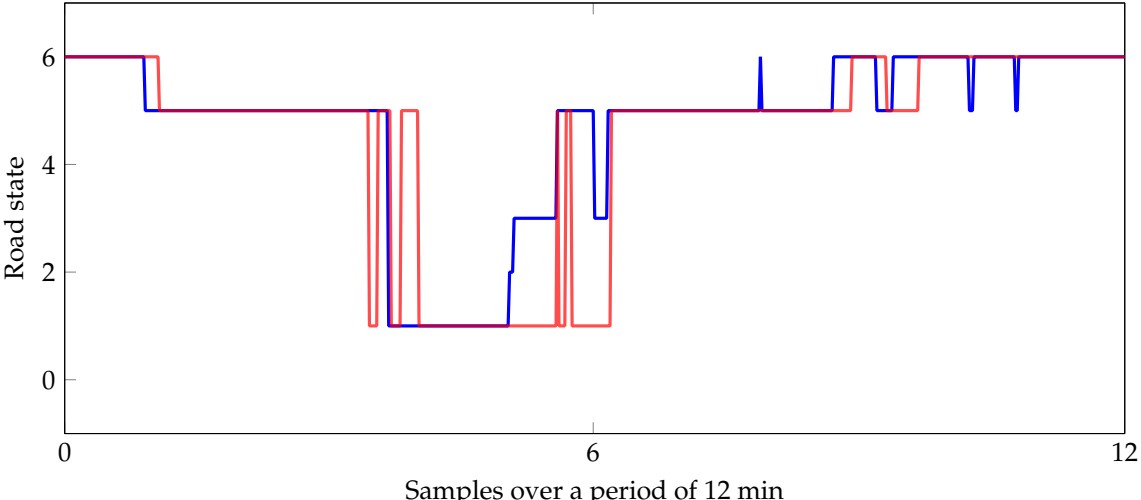

**Figure 5.** RCM411 road state plots from two runs along the same test route: blue, run 1; red, run 2.

## 4.2. MARWIS

This section describes the sensor variables available from the MARWIS sensor, gives a discussion on the reliability of the sensor based on our experience, and comments on collection and storage of the data. In addition, a comparison between MARWIS and RCM411 friction is made.

### 4.2.1. Sensor Variables

The MARWIS sensor offers many of the same variables as RCM411, and also some additional information, relevant in winter road assessment. The variables equal to RCM411 are: friction, road state, air temperature, surface temperature, and water layer thickness. For the road state from MARWIS, a number in the range 1–8 is given where 1 = dry, 2 = damp, 3 = wet, 4 = ice, 5 = snow + ice, 6 = chemically wet, 7 = water + ice, and 8 = snow. The additional variables that MARWIS has are ice percentage, dew point temperature, and relative humidity (air and ground). Three examples of possible data can be seen in Figure 6.

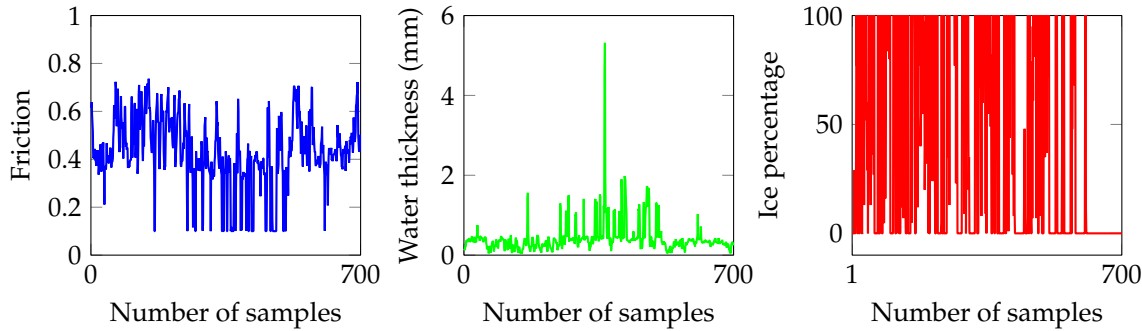

**Figure 6.** Three of the possible data types that can be extracted from the MARWIS: (**left**) friction; (**middle**) water thickness; and (**right**) ice percentage.

### 4.2.2. Reliability

In the initial study, we recognized problems with MARWIS almost immediately. Initially, we had problems with MARWIS losing power, which was caused by an unstable/broken vehicle power adapter. In addition, some issues where identified with the Bluetooth connection between MARWIS and our smart phone, which caused a disconnect between the two, and the application had to restart before further use. This issue was caused by the application itself, and related to the loss of power. Further, the loss of power disrupted the Bluetooth connection to the sensors, which again led to

the phone application crashing without any warnings or reconnects. Another problem included the powering of MARWIS where a converter that came with the sensor converted 12 V to 24 V from the car adapter, which MARWIS required under colder conditions (below a certain temperature, the sensor needs to heat up in order to work properly). The converter did not have any type of safety to limit the voltage, and, when the car was accelerating, the voltage from the adapter increased by a small amount, which led to a higher voltage than MARWIS was made for, frying one of the internal circuits.

### 4.2.3. Data Collection and Storage

The data collection and storage required a stable connection with no application crashes, and the initial reliability tests led to an evaluation in whether to continue with the commercial application. We had earlier discussed how to save the data for further analysis, and the remote server data upload was discarded since raw data were not available. The other option was to use the configuration tool and save locally, but the format of the data in xml was only byte values, which we agreed was not suitable for our purpose. Since we had no idea if the remote server would let us download data in a useful format, we created our own prototype application for communicating with MARWIS and storing the data. Doing research on the communication protocol, communication settings (Bluetooth) led to specific queries with a bytestream checksum directly to the MARWIS sensor. We got a working application which received data from the sensor based on our query and we identified some of the key problems which we experienced in the initial setup. One reason for the crashes in the commercial application was caused by variables not being initialized after a query, which again was caused by either delays in a response or malfunction in the power/bluetooth connection. Seeing that we now had control of the data, we were able to save them in a useful format on the internal storage on the phone for further analysis.

### 4.2.4. Comparison

The initial test results from the MARWIS sensor was extracted on a test-run together with the RCM411 sensor. Thus, since MARWIS and RCM411 have many equal measurement variables, and they are both optical sensors, it was decided that comparing the two would be interesting in the initial phase. The variable that was compared was surface friction, because this is a key variable in winter road assessment. In Figure 7, the graphs of the two frictions plotted, and shows the data collected during a 1 h drive.

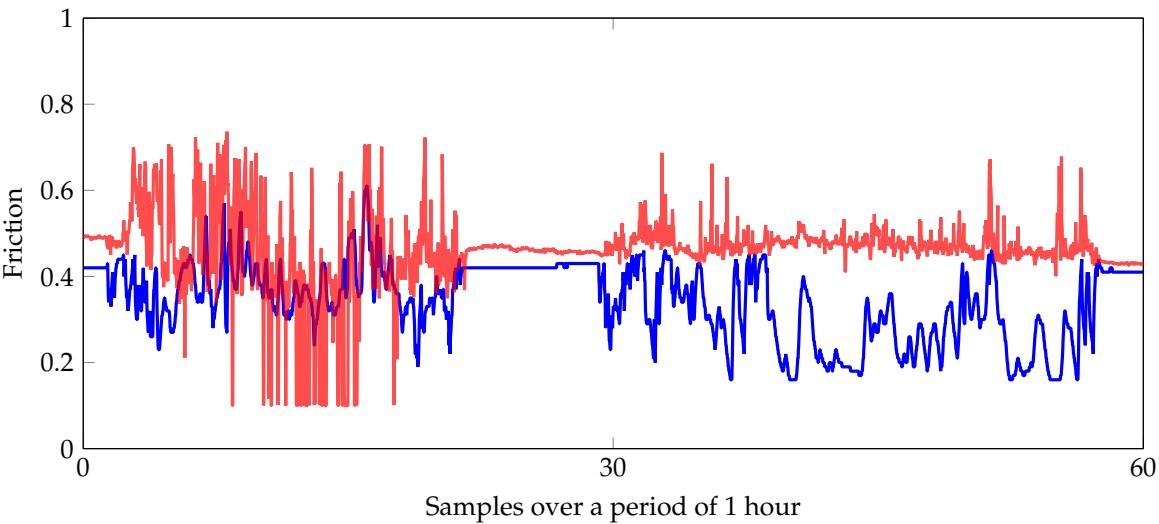

**Figure 7.** The plots of friction data from RCM411 in blue and MARWIS in red, during a 1 h drive.

In the figure, it can be seen that the two sensors show different friction coefficients for almost the entire test drive. From the setup of the sensors, we know that they do not measure the exact same spot, and that the area measured is larger for the MARWIS sensor than the RCM411 sensor. However, the expectations from our part, before analyzing the data, was that areas of equal or similar friction would be observed. Since this is not the case, it would be interesting to do more testing between these two sensors, and compare the results to previous work and comparisons connected to them.

*4.3. Walabot*

This section describes the sensor variables available from the Walabot sensor, and indicates which variables we found relevant. In addition, the section gives a discussion on the reliability of the sensor based on our experience, comments on collection and storage of the data, and shows a first comparison of Walabot image energy against RCM411 friction.

### 4.3.1. Sensor Variables

The Walabot sensor is based on imaging data and can offer several types of output variables. The types are: 3D image, 2D image slice, raw signals, image energy (sum of raw signals), and imaging targets (a list of and the number of identified targets). Figure 8 shows three examples of possible outputs from the Walabot.

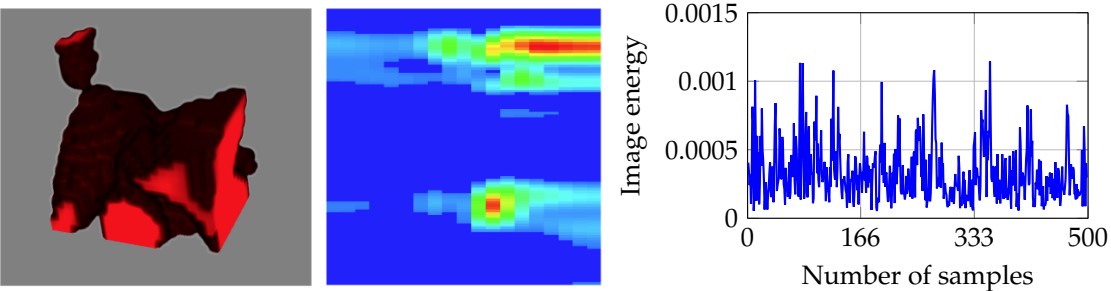

**Figure 8.** Three of the possible data types that can be extracted from the Walabot: (**left**) 3D image; (**middle**) 2D image slice; and (**right**) image energy.

In regards to using Walabot data in winter road assessment, the most suitable data output, having both storage and visual analysis in mind, was predicted to be image energy, since it gives one value every timestep, making it easy to create graphs that can be compared with the other sensor variables.

### 4.3.2. Reliability

Overall, the reliability of the Walabot sensor was adequate. After it was attached to the vehicle, and up and running, no problems occurred when performing the measurements, and data were collected without interruptions. Although collecting data from the initial test-runs went smoothly, the preparations of the Walabot leading up to the test-runs presented some challenges. Firstly, the connection between the Walabot and the Raspberry Pi, via USB, had a tendency to be lost due to a sensitivity in the connection on the Walabot side, where the USB plug needed to be kept stable to secure a connection. This was solved by firmly securing the plug in the Walabot so that it had not room to move. Secondly, the Walabot does not have its own clock, and is therefore dependent on the clock of the device it is connected to. This caused some early headaches seeing that the Raspberry Pi, which the Walabot was connected to, only has a reliable clock as long as it has Internet connection. Thus, when the Raspberry Pi is without Internet access, the clock starts counting from the time it had when the device was last turned off. The strategy used to solve the clock challenge was by either setting up a hotspot on a mobile phone or by manually setting the time on the Raspberry Pi before starting the Walabot.

### 4.3.3. Data Collection and Storage

The collection and storage method used for the Walabot worked well. Collecting the data in multiple files proved to be necessary given that on some test-runs small interruptions occurred leading to the loss of all data in the file that the Raspberry Pi was saving to in that moment. Since the Walabot gave 4–5 measurements every second a large amount of data was collected when the test-runs lasted an hour or more, thus the method of assembling all the data into one file may not always be desirable. In our case, with the initial tests, putting the data into one file seemed like the best choice, in regards to simplifying analysis and comparisons later on.

### 4.3.4. A First Comparison

To get a first impression on how the measurements from the Walabot behave when measuring winter roads, an experiment was conducted with the Walabot together with the RCM411. Both sensors were taken for a test drive and a segment over a period of 20 min was extracted from the sensors and compared. The plots of the two measurements can be seen in Figure 9, where the blue graph is friction from the RCM411, and the red graph is image pixel energy from the Walabot.

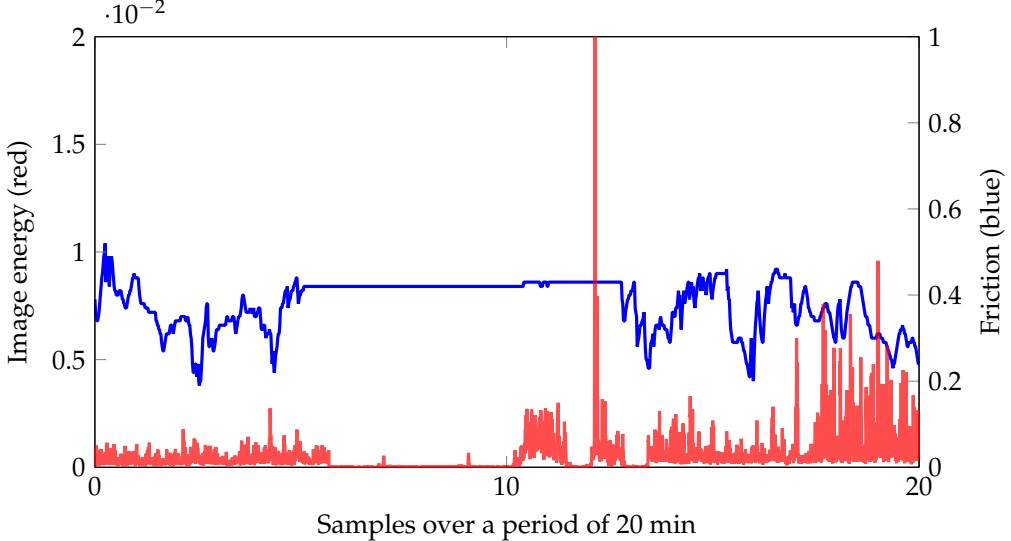

**Figure 9.** The plots of friction data from RCM411 in blue, and Walabot image energy in red, during a 20 min drive.

When comparing the two graphs in Figure 9, it can be observed that in the areas where the friction is close to constant, the image energy has the same behavior giving constant values close to zero, and, for the areas where the friction fluctuates, the image energy also fluctuates. As a first impression, this is a positive result showing that the Walabot radar may have potential in winter road assessment since it seems to have a correlation to, at least, road surface friction. The results from this initial test give strong reasons, and increase the interest, for further investigation of the experimental Walabot sensor for road assessment purposes.

### 4.4. Video, Sound, OBD-II and Smartphone

The measurements and captures collected from the devices/sensors in this section were not the main priority during the startup, and we had no specific goal or ideas of what to expect from the data. Since the measurement setup was mostly based on curiosity and future ideas, no analysis regarding the data captured from the devices/sensors was performed. However, the preliminary experiences are discussed.

### 4.4.1. Video

One of the ideas was to use video capture from in front of the car as a reference for what to expect from the friction measurements behind the car. Another interesting aspect here for future notice was to combine video with artificial intelligence (AI) and predict the road condition (friction) during the video analysis. An example is shown in Figure 10. Capturing this amount of video for an entire trip (several hours) requires both storage capacities on the capturing device as well as storage for further use. Because we were using Box for storage purposes on the other measurement data, we included the video files as well in a folder. With later experiences, this was not a good idea with respect to the capabilities of Box and large file sizes.

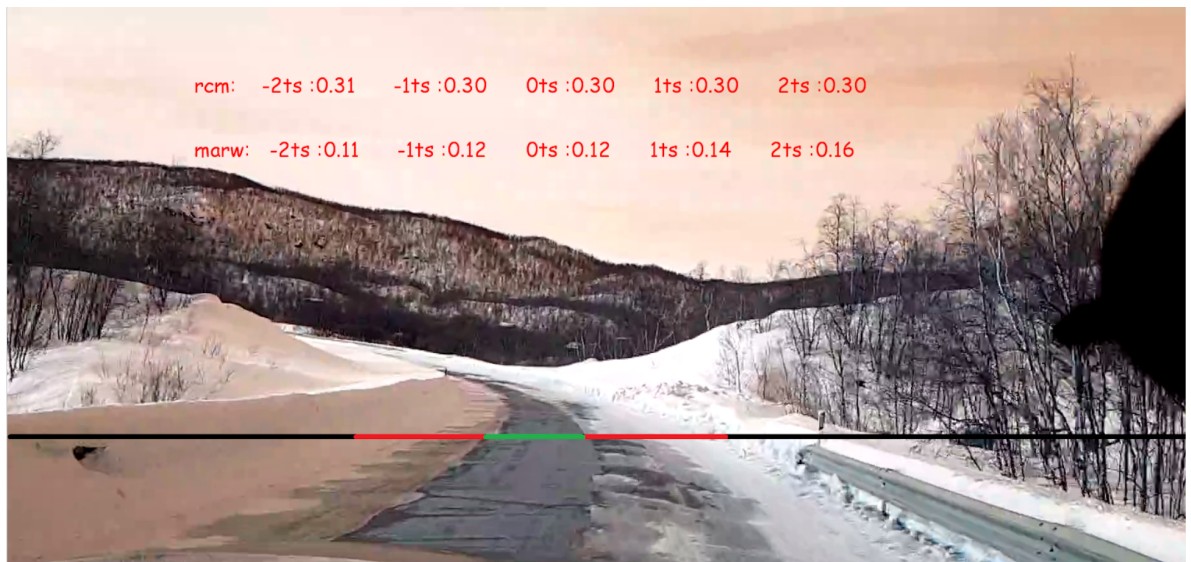

**Figure 10.** Example image on how video can be preprocessed using AI to estimate road conditions. Figure by Kristoffer Tangrand.

### 4.4.2. Sound

We were also doing some experimental work regarding sound capture using a microphone. The idea was to attach a microphone to the car chassis, preferably something as tough as steel to get more accurate vibrations from the ground traveling through the car frame. There were only some initial sound captures and no post processing or analysis of the data have been undertaken yet. The file sizes were so small that they are negligible when uploading.

### 4.4.3. OBD-II

Of the experimental sensors and devices, the OBD-II is already being used for car performance analysis and research purposes. The idea here was to extract information about the car, which is already available, and utilize post processing tools to analyze the data with regards to other measurements such as friction. There are many variables available in the application, where the user can select which ones to record the output from. The available variables differ depending on car model and make, and for our use we selected those we found relevant, both for comparison against other sensors and also analysis purposes.

### 4.4.4. Smartphone

A smartphone is packed with sensors which can record data such as acceleration, orientation, and sound. The commercial application we used for RCM411 had some experimental friction calculations based on measured forces on the phone, that is, using accelerometer to estimate friction and breaking distance. This specific function was optional, and could be uploaded in the same file as the other variables, for comparison against the measured values. Based on these experimental calculations, we

wanted to do our own experiments using smartphone sensors only, and no external information. There were some initial experiments, but this requires more work before doing any analysis on the data.

### 4.5. Hybrid Sensor Technology

In Table 2, a brief summary of the previous results is displayed, based on the criteria from Table 1 together with the price range of each sensor.

**Table 2.** Results of the criteria evaluation described in Table 1, along with the price range (PR) of each sensor.

| Criteria | Sensors | | | | | | |
|---|---|---|---|---|---|---|---|
| | **RCM411** | **MARWIS** | **Walabot** | **Video** | **Sound** | **OBD-II** | **Smartphone** |
| **S1** | 5 | 8 (9) | 4 | 1 | 1 | Optional | 3 |
| **S2** | High | High | Exp | High | Exp | Medium | Medium |
| **R1** | High | Medium | Low | High | High | High | High |
| **R2** | High | Medium | High | High | High | High | High |
| **R3** | High | Low (in-house: High) | Medium | High | High | High | High |
| **D1** | csv | xml (in-house: csv) | csv | 3gp | 3gp | csv | txt |
| **D2** | On stop | Yes | Yes | No | No | On stop | Yes |
| **D3** | Online | Both | Both | Both | Both | Both | Local |
| **T1** | Consistent | Dissimilar to RCM411 | Interesting results against RCM411 | N/A | N/A | N/A | N/A |
| **PR** | High | High | Low | Low | Low | Low | Low |

If we consider the previously discussed sensors in relation to creating a hybrid sensor in the future, there are some interesting observations already. The two optical sensors, RCM411 and MARWIS, have many of the same measured variables so in theory only one of them should be necessary in a hybrid sensor. However, from the friction plots in Figure 7 of RCM411 and MARWIS, an observation was made that the measured friction differed for most of the test drive. This opens up a possibility that the two sensors may be sensitive to different types of conditions, and, if so, a combination of both sensors will be beneficial in a hybrid solution.

The experimental Walabot radar sensor shows, during the first comparison with RCM411 in Figure 9, that there may be a correlation between RCM411 friction data and Walabot image energy data. This is of interest since our prediction is that the Walabot is less affected by snowy and foggy conditions compared to the RCM411 sensor, which will make it an asset during tough weather conditions. In addition, the size and price range of Walabot is a major advantage in relation to a hybrid sensor.

For the device/sensors discussed in Section 4.4, the overall benefits are their small size and that the technology is available for a reasonable low cost. In addition, winter assessment research has been performed on all of them with positive results. The greatest potential lies, probably, within the use of video in winter road assessment, because it brings a new dimension to the analysis that can involve other research areas. Above we presented an example on combining video with AI to predict friction measurements, but the possibilities are numerous. Thus, in our opinion, video will make a valuable inclusion in hybrid assessment. The OBD-II collects data from the vehicle, in contrast to RCM411, MARWIS, Walabot, and video that collects road data. This in itself makes it an important supplementary in the hybrid measurements. Vehicle data can increase the reliability of the risk assessment, since vehicles may act differently under difficult road conditions. Regarding the two last measurements (sound and smartphone data), more experiments will have to be performed to find the strengths they may possess that the other sensors lack.

Implementation of the technology on the vehicle (placement of the sensors) is a topic that will have to be investigated more in future works. The placements chosen in this study, as described in

Figure 1, were based on restrictions connected to the sensors, the type of vehicle used, and our opinion based on predictions and ideas. In a hybrid sensor technology, it will be important to consider on which parts of the road measurements should be conducted, and if sensors should be spread out around the vehicle as opposed to gathered in one place. Gathering the sensors in one place can lead to information being lost (e.g., the two wheel tracks can have varying conditions), and spreading them out may require a larger number of sensors (increasing the price). A solution can be to implement sensors that can measure the whole width of the road and then extract condition data for given areas, for instance with a camera, as shown in Figure 10; however, this will enlarge the post-processing work. The pros and cons of various sensor placements should be identified and mapped so that a justified decision can be made based on the chosen hybrid technology.

## 5. Conclusions

Assessment of road conditions in Nordic countries is a necessity due to increasing amounts of transportation and heavy haulers, which enables the need for accurate safety and risk assessment with strong focus on maintenance and estimates on the occurrence of critical conditions. We performed a preliminary study on several sensor technologies with possible connections to hybrid sensor technology. We tested two commercial and one experimental sensors, as well as four supplementary measuring devices or techniques. In addition, a set of classification criteria was developed (see Table 1) and used (see Table 2) for evaluating the different sensors. The main findings can be summarized as follows:

- The difference in friction for RCM411 and MARWIS, on the test drive, opens up the possibility that the two sensors have different sensitivity to various types of conditions.
- There is a possibility that the RCM411 friction and Walabot image energy are correlated, but the data are currently not sufficient to verify this.
- Video can bring a new dimension to road assessment, by combining it with AI and sensor input.
- The OBD-II data are the only tested sensor data that give information about the vehicle instead of the road, which make them interesting in for hybrid technology.

These early studies established connections between technologies that could potentially decrease risk and, hence, increase safety for all road users. The study also focused on experimental technologies to improve and complement already existing winter road assessment devices. Experimental technologies can present cheaper solutions to replace commercial sensors if the measured variables are correlated. The initial results show promising connections between the tested sensors and possible device combinations which would benefit from a hybrid solution. Implementing a form of hybrid sensor technology could both increase the reliability of risk assessment and have a more complete characterization of the road surface. Having a publicly available joint information source which could identify areas with higher risk that should, if possible, be avoided for safety reasons will increase awareness of difficult conditions. The paper shows several experimental measurements and, as mentioned above, some of them need more work in order to find both strengths and weaknesses. There should also be performed a more thorough field test which incorporates all the above-mentioned equipment to establish a complete view of the measurements compared to each other.

**Author Contributions:** Experimental design, A.P. and T.F.B.; MARWIS experimentation and setup, A.P.; Walabot experimentation and setup, T.F.B.; Setup of other sensors (RCM411, OBD-II, video, sound, and smartphone), A.P. and T.F.B.; Data collection, processing, and analysis, A.P. and T.F.B.; and Writing, A.P. and T.F.B. All authors have read and agreed to the published version of the manuscript.

**Funding:** This paper was partially funded by the project "WiRMa—Winter Road Maintenance", ID 20201092, supported by the Interreg VA NORD-program.

**Acknowledgments:** The authors would like to thank Børre Bang for support.

**Conflicts of Interest:** The authors declare no conflict of interest.

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
