# Peer review of "A Study on Hybrid Sensor Technology in Winter Road Assessment"

_safety, 2020_

Round 1

Reviewer 1 Report

The paper has a conceptual character. Different approaches for application of new technologies in sensors for winter road maintenance are discussed. The scientific soundness of the paper is not sophisticated. Nevertheless the paper is well writen and organized. It can be public in curent form.

Author Response

The authors would like to thank the reviewer for the constructive feedback and kind words. 

Reviewer 2 Report

This paper presents some preliminary experiments regarding the use of hybrid sensors in road assessment. The paper’s main goal is to select and test a number of sensors to assess the possibility of using hybrid measurement technology.

It provides a well-written and well-structured analysis with interesting findings.

However, some issues should be addressed before considering publication:

  1. Make sure all initial assumptions in Section 1 are referenced or justified. Background provided is mainly based on the WiRMa project. It could be interesting to better understand the gap covered in the literature by the paper, and how the proposed methods improve results of other approaches.
  2. Regarding methodological aspects:

- Commercial sensors are already selected before the “authors became part of the project”. However, it could be interesting to elaborate on the different available sensors and the implications of the previous choice.

- The selection of experimental sensors is weakly supported, is seems a little bit subjective.

- Please revise the way you present your findings (e.g. lines 135-136, 255-256, 265-266, 269-270) they are too general and vague for a scientific paper.

- Reliability and Consistency tests are explained in a very broad way, without concrete data and using sentences like “the first mistake was made by us” (please rewrite these parts in a more scientific way).

- Comparison between measurements is weak and not clearly supported with data.

  1. Figure 2 and Figure 8: Units and variables are missing. Even if these figures are examples, it is necessary to show what they are measuring.
  2. The validation of the models is weak; the employed tests are not strong enough to support the results.
  3. Results are well presented and interesting, but very primary and simple. Main concerns are related to the paper’s scientific contribution (it sometimes seems like a commercial report).
  4. Please review some grammar issues (e.g. line 38).

Author Response

The authors would like to thank the reviewer for the constructive feedback and kind words. 

The authors have revised the manuscript and commented on each point made by the reviewer.

Please see the attachment for a point-by-point response.

Reviewer 3 Report

The paper discussed the possibility of use of different sensor technology for winter road condition.

  1. This paper is interesting, but the authors should include some clearer discussions about the benefits and comparisons to previous studies.
  2. Definition of road state should be included.
  3. The scientific background of the sensor technology should be given.
  4. It should have several points highlighted in the conclusions.
  5. Implementation of the technology should be discussed.

Author Response

(The authors gave the same response as above.)

Reviewer 4 Report

This paper investigates a very significant topic from winter road safety point of view. The authors decided to compare and analyse different types of hybrid sensors, which could be capable of helping driver decision making under severe winter conditions.

I would recommend only two issues that should be integrated in the paper.

First, it should be stated in the Introduction (or in Conclusion section) that sensors even with the highest quality are only tools for decision support but eventually the safety depends on the drivers' behaviour of perceiving the sensor alerts. One suggested reference in the topic is Farooq, D., Moslem, S., Duleba S.(2019): Evaluation of driver behaviour criteria for evolution of sustainable traffic safety. Sustainability, 11(11) 3142.

Further, in the method section, a table is missing about the criteria applied in the characterizations of the examined sensors. Also, in the results section, a table is needed for summarizing the measured or estimated data of the examined criteria in the case of the different sensors. These two tables would help the reader to overview the performance of all types of sensors.

After the modification of these two issues, I will recommend the paper for publication.

The paper is well written, I could only spot one mistake: in line 106, the correct writing would be: ...should possess are: a low price, ...

Author Response

(The authors gave the same response as above.)

Round 2

Reviewer 2 Report

The authors have adressed the reviewer concerns.

Reviewer 3 Report

No further comments.

Reviewer 4 Report

All of my comments are followed, the revised version is suitable for publication.